# Sedimentary Ancient DNA (sedaDNA) Reveals Fungal Diversity and Environmental Drivers of Community Changes throughout the Holocene in the Present Boreal Lake Lielais Svētiņu (Eastern Latvia)

**DOI:** 10.3390/microorganisms9040719

**Published:** 2021-03-31

**Authors:** Liisi Talas, Normunds Stivrins, Siim Veski, Leho Tedersoo, Veljo Kisand

**Affiliations:** 1Institute of Technology, University of Tartu, Nooruse 1, 50411 Tartu, Estonia; liisi.talas@gmail.com; 2Department of Geography, Faculty of Geography and Earth Sciences, University of Latvia, Jelgavas iela 1, LV-1004 Riga, Latvia; normunds.stivrins@lu.lv; 3Department of Geology, School of Science, Tallinn University of Technology, Ehitajate tee 5, 19086 Tallinn, Estonia; siim.veski@ttu.ee; 4Institute of Ecology and Earth Sciences, University of Tartu, Ravila 14a, 50411 Tartu, Estonia; leho.tedersoo@ut.ee

**Keywords:** lake sediments, fungal biodiversity, sedimentary ancient DNA, ITS, metabarcoding, paleoecology, paleolimnology, paleogenetics, paleoenvironment, environmental drivers

## Abstract

Fungi are ecologically important in several ecosystem processes, yet their community composition, ecophysiological roles, and responses to changing environmental factors in historical sediments are rarely studied. Here we explored ancient fungal DNA from lake Lielais Svētiņu sediment throughout the Holocene (10.5 kyr) using the ITS metabarcoding approach. Our data revealed diverse fungal taxa and smooth community changes during most of the Holocene with rapid changes occurring in the last few millennia. More precisely, plankton parasitic fungi became more diverse from the Late Holocene (2–4 kyr) which could be related to a shift towards a cooler climate. The Latest Holocene (~2 kyr) showed a distinct increase in the richness of plankton parasites, mycorrhizal, and plant pathogenic fungi which can be associated with an increased transfer rate of plant material into the lake and blooms of planktonic organisms influenced by increased, yet moderate, human impact. Thus, major community shifts in plankton parasites and mycorrhizal fungi could be utilized as potential paleo-variables that accompany host-substrate dynamics. Our work demonstrates that fungal aDNA with predicted ecophysiology and host specificity can be employed to reconstruct both aquatic and surrounding terrestrial ecosystems and to estimate the influence of environmental change.

## 1. Introduction

In recent years, a growing number of studies have used lake sedimentary ancient DNA (sedaDNA) to reconstruct past ecosystem changes. Examples include studying the impact of invasive species [1], reconstructing a history of freshwater fisheries [2], and exploring floristic diversity changes [3,4]. However, the majority of these studies on past ecosystems and ecosystem changes have used ancient plant or animal DNA with much less attention paid to ancient fungi [5,6] which continue to be studied using traditional methods such as identification of spores or fossil remains [7,8]. To our knowledge, only a few studies from permafrost [5,6,9], sea sediments [10], lake [11,12,13], and cave sediments [14] have used molecular-based methods to assess the past fungal or whole eukaryotic diversity in historical ecosystems. Some of these studies [5,10,11,12,13] used universal 18S rDNA region primers that enable one to detect a wide variety of eukaryotes but restricts the number of identified fungal taxa. In contrast, Bellemain et al. [6] and Kochkina et al. [9] focused on the internal transcribed spacer (ITS) region that serves as the main DNA barcode marker used to assess fungal diversity [15,16]. ITS can discriminate between closely related species for a broad range of fungi compared to the nuclear ribosomal small subunit (SSU), which has poor species-level identification features in fungi.

Fungi are a highly diverse kingdom of eukaryotic organisms that play important ecological roles as saprobes, symbionts, or parasites of plants and animals. Therefore, changes in fungal communities over time indicate changes in the entire ecosystem and enable one to untangle the ecological status of the environment. Fungal richness in terrestrial environments, such as soil and plant-related habitats, is highly diverse [17]. Their spores and mycelia often transfer into lakes by soil particles, rain, wind, or together with other organic matter such as dung or plant material [18]. A recent study [11] showed that sedimentary aDNA, especially from fungi and plants, is useful for the reconstruction of the environment surrounding the lake. However, compared to terrestrial fungi, information about the biodiversity of aquatic fungi is still poorly documented. Nevertheless, fungi are common residents in aquatic habitats and represent one of the main groups of microbial eukaryote communities in deep-sea and lake sediments [10,11,19] where they play crucial roles in the turnover of organic matter. For example, parasitic chytrids act as important mediators in aquatic food webs and transfer nutrients from inedible phytoplankton to zooplankton (“mycoloop”) [20].

Here we used aDNA from lake sediments to assess the composition of past fungal communities, their functional roles, and community changes throughout the Holocene (in the last 10.5 kyr) in relation to environmental factors. We demonstrate that following a broad range of functional ecological roles of fungi offers a good proxy for interpretation of overall past biodiversity changes beyond fungal diversity. We hypothesize that the abundance and diversity of fungi with broad functionality (e.g., saprotrophs) and broad host range has been more stable over the Holocene while the diversity of fungi with narrow specificity of substrate or hosts has been more variable following the dynamics of host or specific substrate. To test our hypothesis and to achieve the above-mentioned general aims of this study, we: (a) re-constructed the past fungal community composition and their ecological diversity from lake sediments, (b) tested the predictive value of fungal diversity as a proxy for broad ecosystem changes, and (c) explored how ancient fungal communities responded to changes in climatic and environmental factors during the Holocene.

## 2. Materials and Methods

### 2.1. Sampling and Chronology of the Sediment Cores

The lake Lielais Svētiņu (LSv) is an important study site since it lies in a climatic and vegetative transition area and spans both the maritime-continentality line that runs west to east from the central Baltic area to NE Europe and transects the gradual decay of nemoral forest into boreal. Lielais Svētiņu has a long sedimentary record and well-characterized relatively late human impact, thus making it a useful site to decipher environmental changes in this climate zone (see Section A.1).

A sediment core was taken for aDNA analysis from the middle of the lake Lielais Svētiņu (LSv) (eastern Latvia; 56°45′ N, 27°08′ E, 4 m water depth; Figure 1) in March 2013 using a Russian type corer (1 m long, diameter of 10 cm). In total, an 11 m thick sediment was obtained. Independent samples (layers) were collected at 2.5 cm intervals from cores in the Institute of Geology in Tallinn, Estonia. Fifty-five layers (Appendix A) were picked out for further analysis depending on the estimated past temperature changes [21]. Samples were collected more frequently from sediment areas with rapid temperature changes. From each individual layer, 3 biological replicate samples were taken in a row (see Figure A1 under Section A.6). All the subsampling was done using protective lab wear and clean equipment (Section A.2). The chronology of the LSv sediment core is based on radiocarbon dates from the sediment core described in Stivrins et al. [21], which were adjusted with our sediment core by the age-depth model (full description in Kisand et al. [11]).

### 2.2. Molecular Analysis

DNA extraction and amplification were performed in separate labs (buildings) dedicated for aDNA work under a cleaned UV-treated positive-flow hood (Kojair K-safety KR-125, Mänttä-Vippula, Finland) using clean equipment and protective lab wear (details in Section A.3). Total DNA was extracted from sediment samples in 3 biological replicates (total of 162 samples) using PowerSoil^®^ DNA Isolation Kit (MoBio Laboratories, Carlsbad, CA, USA) according to the manufacturer’s protocol. Sediment samples homogenization was conducted using Fast Prep^TM^-24 Tissue Homogenizer (MP Biomedicals) for 55 s at 4 m/s. All extracted samples were stored at −20 °C until PCR amplification.

Metabarcoding libraries were performed in two steps (PCR1 and PCR2). PCR1 amplification targeted internal transcribed spacer 2 (ITS2) region using fungal specific multiplex of 5 forward primers (ITS3NGS1, ITS3NGS2, ITS3NGS3, ITS3NGS4, ITS3NGS5) and reverse primer ITS4NGS [17]. 0.75 μM of forward and reverse primers were added to 20 μL reactions containing 1–1.5 μL aDNA (depending on sediment depth), 2× Phusion Master Mix with HF buffer, 1 μg/μL BSA (Thermo Scientific, Vilnius, Lithuania), and ultrapure water (MoBio Laboratories, Carlsbad, CA, USA). Phusion High-Fidelity DNA polymerase (Thermo Scientific) was used for amplification program containing the following steps: denaturation at 98 °C for 30 s, followed by 30 cycles of 10 s at 98 °C, annealing at 46 °C for 30 s, and extension at 72 °C for 15 s, with a final extension at 72 °C for 10 min. The presence and quantity of amplicons were visualized using 1.4% agarose gels (1xTAE) containing ethidium bromide. All samples without visible amplicons on the gel were amplified three times, mixed, and purified/concentrated using Select-A-Size DNA Clean and ConcentratorTM kit (Zymo Research, The Epigenetics Company). Technical replicates (11 samples were amplified and sequenced twice) were used to test PCR and sequencing consistency. PCR2 was performed to tag PCR1 amplicons with Illumina TrueSeq adapters and P5/P7 tail indexing primers. Step 2 PCR reaction (20 μL) contained 2× Phusion Master Mix High-Fidelity buffer (Thermo Scientific), 25 μM Mplex primer, 2.5 μL long PCR index X, 1 μL of PCR amplicons, and nuclear-free water (Thermo Scientific). The Step 2 amplification followed the program: 2 min at 98 °C, followed by 12 cycles of 20 s at 98 °C, 30 s at 65 °C, 30 s at 72 °C; and final extension 5 min at 72 °C. In each step, negative controls (no added DNA) were used to avoid possible cross-contamination from reagents or the environment. Amplicons were sent to Illumina MiSeq 2 × 250 bp sequencing at the FIMM (Institute of Molecular Microbial Medicine Finland, Helsinki, Finland).

### 2.3. Bioinformatics Analysis

Raw reads were quality-trimmed using Trimmomatic (v. 0.32) (average quality score <30Q). The VSEARCH tool [22] was used for further reads pairing, dereplication, chimera removal, sorting, <4 read clusters removal, molecular operational taxonomic unit (mOTU) clustering at 97% similarity threshold (see Section A.4 for a detailed bioinformatics description). Additional post-clustering processing was conducted using the R-package LULU [23], which reduces erroneous mOTUs. The mOTUs taxonomic affiliation was obtained using the fungal reference database UNITE (UNITE ver. 7; [24]). We relied on 98%, 90%, 85%, 80%, and 75% sequence identity as a criterion for assigning OTUs to species, genus, family, order, or class level, respectively [17]. All non-fungal OTUs (e.g., *Bacteria* and *Plantae*) were discarded from further analyses. The sequence data has been submitted to the SRA database under BioProject PRJNA704193.

### 2.4. Assigning Ecological Roles of Fungi

The ecological roles of the fungi were cross-validated using various tools and composed to determine both the possible relationships between identified fungi and their hosts and identify fungal groups that are responsive to change in environmental factors. A number of tools/databases were leveraged for this purpose, including FUNGuild [25], FungalTraits [26], Plant Parasites of Europe [27], Fungal Families of the World [28], among others (Appendix A). For each identified fungal mOTU, we assigned a trophic status, lifestyle, environment, mycorrhizal associations, and host-specificity.

### 2.5. Statistical Analysis

All statistical analyses were conducted using R version 3.5.1 [29]. Details about technical and biological replicates can be found in Section A.5. For downstream analysis, the technical replicates were excluded, and biological replicates were merged together (see Section A.5, Appendix A).

#### 2.5.1. Richness Changes between Time Periods

We used a *Venn* diagram (InteractiVenn; [30]) to visualize shared mOTU richness between four time periods: Latest Holocene (last 2.0 kyr), Late Holocene (2.0–4.0 kyr), Mid-Holocene (4.0–8.0 kyr), and Early Holocene (8.0–10.5 kyr). Principal component analysis (PCoA) (*vegan* package) was conducted to visualize the dynamics of the abundance of various mOTUs over time. In addition, we employed the PERMANOVA (*adonis* function in *vegan* R package; [31]) on mOTU normalized counts to test changes in the community between time periods.

#### 2.5.2. Community Composition and Ecology of Fungi

We employed the *codyn* package [32] to measure community composition changes over the 10.5 kyr study period. For that we used the functions *rate_change()*, *variance_ratio()*, *synchrony()*, and *turnover()*. The *rate change* shows the rate of directional change in community composition over the study period using Euclidean distance [33]. The *variance ratio* and *synchrony* characterize patterns of species covariance and synchronous/asynchronous fluctuations. The *variance ratio* ranges from <1 (negative covariance) to >1 (positive covariance) and is centered at 1 (species vary independently). To detect asynchrony between species, we used *“gross” synchrony* metrics whose values range from −1 (complete asynchrony) to 1 (complete synchrony) and are centered at the value 0 which indicates that species fluctuate independently [34]. The *turnover* calculates the total species turnover metric, which shows the proportion of different species between two-time points [32].

#### 2.5.3. Past Environmental Drivers

To test the relationship between past environmental factors and the richness of different fungal groups, we used the following paleoecological proxies: summer temperatures (T_sum_), continental climate (T_cont_), waterlogging (W_tol_), drought (D_tol_), and shade tolerance (S_tol_), the openness of surrounding area (OPEN), concentration of charcoal particles (Ch), pyrite (abundant in anoxic sediment, FeS_2_) and human-related pollen (HRP) [21]. T_sum_, T_cont_, OPEN, W_tol_, D_tol_, S_tol_, and HRP were reconstructed based on pollen records taken from the 2009 core [21]. Continental climate (T_cont_) values were calculated as T_sum_–T_win_. Landscape openness (OPEN) was based on the pollen of dominant herbaceous taxa in the lake surrounding. The tolerance values of 14 trees/shrubs were used to reconstruct W_tol_, D_tol_, and S_tol_. Also, Charcoal particles (Ch) in the sediment were used as indications of fire events and pyrite (FeS_2_) as an indicator of anoxic conditions in the lake. HRP reconstruction was based on cereal species (*Secale cereale*, *Hordeum vulgare*, *Triticum aestivum*, and *Avena sativa*) that have been related to agricultural activities. The sediment core taken by Stivrins et al. [21] in 2009 from the same location can be compared with the sediment core used in this study. The data from two cores (2009 and 2013, this study) was parallelized using the age-depth model [11].

Raw reads data from different fungal groups were normalized (cumulative sums scaling—CSS, package *metagenomeSeq*, [35]), merged among biological replicates, and used for smoothing. mOTU data were combined among replicates and used for smoothing. We used the *gam* function from the *mgcv* package to fit the Generalized Additive Model (GAM) on nonlinear relationships. Smoothness estimation for the model was conducted using cubic splines (bs = “cr”), where k (knots) was chosen depending on the model fitting (the majority have 10 knots per 55 time points). GAM models were used to establish overall richness changes in fungal groups and to reduce variance over time (see Section A.5). We used fitted values from the model for two-sided Pearson correlation (*cor.test* function) with a confidence level of 0.95, which was used to study possible interactions between fungal groups and paleoecological proxies. Correlations were identified by correlation rate as very strong (>0.8 or <−0.8), strong (>0.6 or <−0.6), moderate (>0.45 or <−0.45) and weak (<0.45 or >−0.45) at *p* < 0.05.

## 3. Results

### 3.1. Fungal Sequences

Sequencing of the ITS2 region resulted in 53,006,690 raw read pairs of 55 triplicate samples (Appendix A). After quality-filtering with Trimmomatic, 48,035,087 read pairs (90.6%) survived and were merged into 19,862,567 full-length reads. These were dereplicated into 1,539,800 identical sequence groups and all the reads <200 nt long were discarded from the dataset, leaving 19,091,590 reads. Further data denoising from chimeras and rare sequences (<4 reads) resulted in 16,361,004 reads. These sequences were clustered into 2708 molecular operational taxonomic units (mOTUs, <97% similarity). In addition, the post-clustering method LULU removed 20 mOTUs. BLASTn analysis gave no match to 1340 mOTUs and identified 115 non-fungal taxa (*Plantae*, *Protista*, and *Protozoa*), which were removed from downstream analyses. Out of 1233 fungal mOTUs (11,387,536 reads), 391 (32% of mOTUs) were determined as unidentified Fungi. In addition, we measured DNA fragmentation in the sediment samples using a DNA analyzer (TapeStation 2200) which showed that 300–350 bp long fragment amplification is possible even from deep sediment layers (Appendix A). For further statistical analysis, the technical replicates were removed. Unique mOTUs were combined and raw reads merged between biological replicates of each layer, forming a total of 55 samples with 1125 unique mOTUs (Appendix A).

### 3.2. Overall Phylogenetic and Ecophysiological Richness of Fungi

Nearly 2/3 of mOTUs (68%) were classified at the Class level of taxonomy or lower (Figure 2A). In total, the fungal richness was greatest in phyla *Ascomycota* (30%) and *Basidiomycota* (27%), followed by a smaller proportion of mOTUs in *Chytridiomycota* (4%), *Rozellomycota* (6%), *Zygomycota* (2%), *Glomeromycota* (0.1%). The highest richness was recovered in the classes *Agaricomycetes* (96 genera), *Sordariomycetes* (35 genera), and *Dothideomycetes* (32 genera) (Appendix A). *Chytridiomycota* was represented by the classes *Chytridiomycetes* and *Monoblepharidomycetes*, yet most of these (67%), and all fungal mOTUs in *Rozellomycota*, remained unclassified at lower taxonomic levels. Most of the mOTUs (997 OTUs) were found in a small number of layers (1 to 5 layers; median = 1), while 61 mOTUs were detected in ≥10 layers (total of 55 layers) serving as a core community. The most frequent mOTUs throughout the sediment (in >40 layers) were fungi in the genera *Cladosporium*, *Saccharomyces*, and *Ganoderma*, which include mainly plant pathogens and saprotrophs. Further, many other aquatic and terrestrial saprotrophs (*Hyaloraphidium* sp., dung saprotroph *Coprinellus* sp., wood saprotroph *Xylaria* sp., yeast *Candida* sp., and *Cryptococcus* sp.) were observed throughout the sediment (in >30 layers), but not in any control samples. The full taxonomy is presented in Appendix A.

In general, about 40% of the classified mOTUs belong to fungi that are expected to be terrestrial, while 23% were determined as aquatic and 12% as inhabiting both environments, leaving 25% of fungal habitats classified as unknown (Figure 3A). The most numerous guilds were terrestrial saprotrophs (294 mOTUs; 26%) and pathotrophs (245 mOTUs; 22%), in particular, wood and litter saprotrophs (128 mOTUs), as well as pathogens of plants (205 mOTUs), animals (140 mOTUs; including insects), and other fungi (61 mOTUs). Furthermore, we determined 40 mOTUs related to specific plant taxa and 23 mOTUs characteristic of certain forest types (conifer vs. deciduous trees) (see Appendix A). Symbiotrophs were less numerous (41 mOTUs; 3.6%), including lichens and mycorrhizal fungi (arbuscular, ericoid, and ectomycorrhizal fungi) (Figure 3B). Among fungi with aquatic habitats, the most numerous were parasites of plankton (phytoplankton and other planktonic organisms) (223 mOTUs). In addition, we were able to detect a small number of fish pathogenic fungi (12 mOTUs). However, multiple ecological roles were assigned to 25.4% of mOTUs with the same identity and no ecological traits to 23% of mOTUs due to insufficient similarity to their described species or genera (see Section A.7).

### 3.3. Dynamics and Development of the Fungal Community

We conducted PCoA analysis and linear fitting of age and the concentration of human-related pollen (HRP) to visualize the dynamic changes in the abundance of mOTUs over time (Figure 2B and Appendix A). We see a smooth change in the community during the Holocene with more discontinuous and rapid changes during the Latest Holocene which distinctively contrasts from the rest of the samples. These results are also supported by a PERMANOVA analysis that shows significant differences between time periods (Appendix A). We observed a three-fold increase in mOTUs during the Latest Holocene period (last 2 kyr, 37% of mOTUs, 2 new classes, Appendix A). Analysis of the shared mOTUs between time periods supports a model of gradual change in the community from the Early to Late Holocene (11 kyr to 2 kyr), while the Latest Holocene distinctively differed (Appendix A). An increasing number of mOTUs were shared between consecutive time periods: 10% between Early and Mid-Holocene; 13% between Mid-Holocene and Late Holocene; 18% between Late and Latest Holocene).

To measure changes in the community composition over time, we used temporal diversity indices (*rate change* and *turnover*) and community stability metrics (*variance ratio* and *synchrony*) ([32]; Table 1, Figure 4). We observed a high total species turnover in the majority of fungal groups (mean ~80%) between adjacent time points (Figure 4) due to many appearing and disappearing species. Yet, the composition of fungal mOTUs that originate from terrestrial habitats was rather stable all over the 10.5 kyr (Table 1, Appendix A), which shows the importance of reappearing species. The highest rate of directional change in a community (*rate change;* [33]) appeared in the groups of aquatic fungi i.e., plankton parasites and in mycorrhizal fungi (Appendix A). In contrast, (especially in plankton parasitic fungi) the proportion of species turnover was gradually lower (Figure 4), illustrating a growing number of appearing species. Likewise, the *variance ratio* was high in these groups, which indicates species covariance over the entire timespan (Table 1). In addition, *“gross” synchrony* [34] showed that ~25% (median = 0.24; min–max: 0.14–0.26) of these species fluctuations depend on each other, thus illustrating that a quarter of species in the community responded in a similar way to environmental changes. In contrast, the highest asynchrony (32%) appeared in lichen symbionts, where species fluctuated opposite to each other. All the other larger ecological groups (terrestrial fungi, plant pathogens, saprotrophs, symbiotrophs) showed a relatively low *rate change* in community composition and *“gross” synchrony* values. Also, the species turnover was lower for saprotrophic fungi (i.e., wood, litter, and dung saprotrophs) (Figure 4).

### 3.4. Past Environmental Drivers

We used Generalized Additive Model (GAM) to denoise the variance in abundance data of fungal ecophysiological groups and the fitted values were related (correlation analysis) with available environmental paleoproxy variables (Figure 5 and Appendix A). Two-sided Pearson correlations between abundance in fungal ecophysiological groups and environmental paleoproxies illustrate significant positive correlations with continental climate (T_cont_), openness of surrounding area (OPEN), concentration of charcoal particles (Ch), and human-related pollen (HRP), and significant negative correlation with summer temperatures (T_sum_), shade tolerance (S_tol_), and pyrite (FeS_2_) (Figure 6, Appendix A). In addition, plant pathogens also hold strong positive correlations with drought tolerance (D_tol_) (*r* = 0.62) and strong negative correlations with S_tol_ (*r* = −0.66) and T_sum_ (*r* = −0.61). Fish pathogenic fungi with low richness had a strong positive relationship to waterlogging tolerance (W_tol_) (*r* = 0.68), whereas saprotrophs of litter, wood, and dung showed only a moderate correlation with different environmental paleoproxies.

## 4. Discussion

We demonstrated that fungal aDNA is well preserved in lake sediment shown to have a good record of various paleo-environmental and ecological materials including pollen, micro- and macrofossils [7,36]. As with the archived material analyzed by traditional paleoecological methods, fungal aDNA originates from both terrestrial and aquatic habitats and mirrors the community composition in the lake ecosystem and watershed. Therefore, when the ecophysiology, including the host specificity of fungal species archived in the lake sediment as aDNA, can be predicted, this information can be employed to reconstruct both the aquatic ecosystem and the ecosystem in the surrounding terrestrial landscape. Earlier, non-aDNA-based studies have demonstrated gradual ecosystem changes in this lake and its surroundings [21,36], driven by gradual climate changes during the Holocene. At the same time, an abundance of fungi with broad ecophysiology (e.g., saprotrophs) were relatively stable over the ~10.5 kyr study period and thus, more robust to environmental changes than narrow-substrate and host-specific fungi, such as pathotrophs (e.g., plankton parasites) or symbiotrophs [26,37,38], which follow the host and/or substrate dynamics. Contrastingly, more rapid ecosystem changes occurred during the last two millennia with a slow but steady increase in the anthropogenic influence in the area. Characteristic to this phenomenon is higher fungal richness among pathogenic fungi (plankton parasites, plant pathogens) and mycorrhizal fungi. Increased human activity is associated with an increased level of forest fires, floristic diversity, and landscape openness [36,39]. These environmental drivers have a different nature when compared to natural fluctuations in the structure of vegetation described by paleoproxies such as waterlogging, drought, and shade tolerance. In addition, although terrestrial and aquatic fungal communities were affected by the same environmental drivers, the more prevalent community changes appeared in aquatic fungi, possibly indicating a susceptibility of aquatic organisms to adjust to modest environmental disturbances.

### 4.1. Fungal Community Composition and Their Ecological Diversity

The general proportions in the fungal taxonomic affiliations at the phylum level (Figure 2A) were compared to previous studies from permafrost [6] and aquatic environments [18,19,40]. In contrast with sea sediments, we observed members of *Glomeromycota* and a higher proportion of *Basidiomycota*, which is more similar to terrestrial environments in this region [41]. *Agaricales* and *Polyporales* were the prevalent *Basidiomycota* orders (Appendix A). The species in these orders are mostly litter or wood saprotrophs, plant pathogens, or ectomycorrhizal symbionts (*Agaricales*) and likely represent transient species of terrestrial origin that have been carried to the lake with organic matter or spores. Thus, the high richness of terrestrial fungi preserved in lake sediments in all layers studied (Figure 3A), confirms the value of these archives to reconstruct the surrounding past terrestrial biodiversity [11], especially through interactions with their host species [7]. The high total fungal richness and successful species- and genus-level identification enabled the detection of host-specific fungi (46 mOTUs) (e.g., the plant pathogens of *Alnus*, *Betula*, *Salix, Frangula*, *Picea*, *Poaceae*, or mycorrhizal fungi of *Pinus* and *Salix* roots). The host-specific fungi offer valuable information about host populations in the past because fungi likely leave a stronger signal (higher concentration of DNA) within the sediment compared to host DNA (e.g., mammal DNA; [42]) due to their DNA being both widespread and protected by spore persistence in the sediment. Therefore, fungal host-specificity or associations to organism groups would enable us to use them as signatures, that could be used to study the appearance and dynamics of plant diseases in paleoenvironments.

In addition to terrestrial fungi, we also observed many *Ascomycota*, *Chytridiomycota*, and *Rozellomycota* characteristics of aquatic environments [19,43,44]. These fungi are known to play important ecological roles in aquatic habitats as saprobes on a variety of substrates or as parasites on phytoplankton, zooplankton, other fungi, or even aquatic invertebrates [45,46]. As a result, the presence of species from all 4 orders of *Chytridiomycota* (phytoplankton parasitic *Rhizophydiales* and *Lobulomycetales*, saprotrophic *Monoblepharidales and Chytridiales*) were observed in our samples. Also, multiple studies have shown *Rozellomycota*, which infects various organisms including large diatoms [44,47] and smaller planktonic eukaryotes [46,48].

### 4.2. Fungal Community Changes and Paleoecological Drivers

In conjunction with changes in climate and the dynamics of ecosystems (e.g., average temperatures, vegetation proxies in the region) we observed a gradual change in the fungal community during most of the Holocene (Figure 2B and Appendix A). Possibly due to the low number of mOTUs in the core community and relatively long time intervals obtained by subsampling the cores with a few centimeter intervals, the turnover at the species level was high (~80%) in most groups between consecutive time points (Figure 4). However, a low rate of community change was observed in most ecophysiological groups over the ~10.5 kyr (Appendix A). For example, we observed that terrestrial fungi (including about 40% of all mOTUs) had a relatively stable community over the entire timespan (Table 1, Figure 3A and Appendix A), exhibiting 3 times lower rate of community change than in aquatic fungi. Therefore, the aquatic fungi could be more susceptible to changes in environmental factors than terrestrial groups. Similarly, in the ecological groups of saprotrophs, animal, and fungal pathogens, we did not observe any major community changes over the Holocene (Table 1, Appendix A) and the proportion of species turnover was lowest for saprotrophic fungi (e.g., dung, wood and litter saprotrophs). All of these changes in richness can be considered natural fluctuations in diversity (Figure 3B). Thus, terrestrial groups of litter, wood, and dung saprotrophs demonstrated only moderate correlations with different environmental paleoproxies that do not induce community change. In addition, we show that fish pathogen fluctuations are correlated to waterlogging (Figure 6A) even though their communities did not drastically change. This may indicate possible variations in fish populations.

We observed that fungal richness accelerated in the last two or three millennia with an increase in the total number of mOTUs by ~37% (Figure 2B, Figure 3 and Appendix A). Most of these mOTUs belong to aquatic fungi (i.e., plankton parasites) with some exceptions. In contrast with the rise in richness, we observe a decrease in species turnover (Figure 4). This corroborates the formation of a community with newly added species and therefore more ecological niches in the lake. Our observation agrees with studies by Stivrins [21] and Kisand [11], who documented an increase in phytoplankton, especially Chlorophyta (green algae), over the last 2.0 kyr in LSv sediments. Phytoplankton species are the primary producers in lakes and zoosporic fungal parasites help to control the extent of their blooms [37]. Phytoplankton parasitic fungi have many possible roles in food web dynamics [45]. One role is contributing to nutrient transfer from inedible phytoplankton to zooplankton (“mycoloop”, [37]). Therefore, any shift in phytoplankton also impacts other aquatic organisms. In our study, the community change in plankton parasitic fungi is likely connected to an increase of phytoplankton or other planktonic organism richness.

In contrast with other terrestrial fungi, the species number (as mOTUs) of plant pathogens but also in groups of conifer-related fungi and mycorrhizal fungi increased during the last two millennia. Interestingly, the higher rate of community change appeared only in plankton parasites and in mycorrhizal fungi groups, where ~25% of plankton parasitic species and ~13% of mycorrhizal species fluctuated dependent on each other and respond similarly to environmental factors (Table 1, Appendix A). Thus, regardless of the rise in richness during the last ~2 kyr, and highly fluctuating species in adjacent timepoints (~60–80%), the communities of plant pathogens and conifer related fungi were rather stable over the entire timespan (Figure 3 and Figure 4, Table 1, Appendix A). Therefore, we conclude that some fungi with narrower host group or substrate specificity (e.g., plankton parasites, pathotrophs, and ectomycorrhizal fungi) could be more responsive to ecosystem changes than wide spectrum species such as saprotrophs or terrestrial fungi in general [37,38].

### 4.3. Increased Human Impact and Change in Richness of Fungi

The immediate surrounding of the lake under study has remained relatively pristine from direct human activity which can be associated with the appearance of HRP which gradually rose over the last 2 kyrs with a distinct increase during the last 700 yrs. Human activities have been shown to positively influence the vegetation richness of LSv [21] that possibly enriched the lake with additional nutrients (e.g., agriculture, grazing) which could explain the increase in richness of plant pathogenic and mycorrhizal fungi over the last 2 kyrs. Also, strong associations with drought indicate stress-related effects where vegetation is more prone to pathogens. In addition, the strong correlations to Ch and OPEN (Figure 6A) could also be results of increasing and widening human activity in the region, especially because Ch and OPEN are not the main factors impacting terrestrial communities in the late to early Holocene (2–10.5 kyr) (Appendix A, Figure 6A,C). Ch is used as evidence of fire events in the past and is suggested to increase with drought [49] and human activities [36] such as slash-and-burn agriculture which alters the landscape. Forest fires are key factors that explain disturbances in forest and landscape structure [39,49] and can at least partly explain the increasing openness of the area (Figure 5A). Open areas enable more widespread transport of plant material to the lake and may reflect the larger-scale events. Therefore, a rapid increase in OPEN and Ch could be the result of wider human impact over the region (>2.0 kyr) whereas temperature, W_tol_, S_tol_, and D_tol_ were mainly driven by natural factors. When excluding the last ~2 kyr the fungal communities were affected by waterlogging, shade, and drought tolerance as natural environmental drivers.

An increase in nutrients likely brought along the rise of phytoplankton and other planktonic organisms that further lead to an increase in the richness of plankton parasites in the lake. However, we also detected an increase in the richness of plankton parasitic fungi in the last ~4 kyr which is earlier than the time we see eukaryotic algae dominance (>2.0 kyr). This is possibly due to a late Holocene climatic-environmental change either driven by a shift in cyanobacteria-eukaryotic algae (eukaryotic algae dominance only appears at lower temperatures, [21]) or possibly due to cooler temperatures favored by parasitic fungi. Chytrid epidemics can appear even on optimal growth conditions for the host [37]. Therefore, the drivers of the observed community changes in plankton parasitic fungi are likely interactions with the shift to a cooler climate during Late Holocene and human activity and landscape openness in the Latest Holocene.

### 4.4. Challenges with Using Fungal Richness and Diversity as a Proxy for Total Paleo-Diversity

One of the first challenges of broad fungal community detection is efficient DNA extraction. Obtaining high yield and purity of aDNA preserved in the sediment is a demanding task due to the effects of humic-acid binding and the presence of clay components [50,51]. This can be achieved via taking extreme care with DNA extraction, PCR bias, possible contamination, and DNA degradation, among other concerns (see Section A.8). Another key challenge is to assign the habitat and ecophysiology to the resulting mOTUs based on a short ITS sequence because the origin of acquired fungal taxa is very broad in our sediment archives. The sequences can represent fungal diversity in sediment biomes, originate from water columns, or terrestrial environments [18]. One of the challenges of determining the fungal origin or their ecophysiology lies in the inability to assign a fungal identity to all detected fungi and unresolved taxonomy, especially for aquatic fungi. Therefore, species detection may be the key to acquiring reliable fungal signatures/proxies for paleo-environmental research. In addition, the habitat generalists may be less usable as a proxy than fungi with narrow ecological roles. Nevertheless, we were able to detect a wide variety of fungal taxa and their ecological roles and were able to reconstruct the past biodiversity. Despite this success, our work would benefit from additional research focused on generating reference sequences for databases and studies on fungal ecology. This additional information would enable us to detect an even more detailed understanding of fungal signatures in the sediment and increase their predictive value in ecosystem changes. Although the multiplex primer approach enables a reliable estimate of fungal richness, moving to a quantitative assessment of fungal biodiversity dynamics is an additional challenge due to high variation in ITS region repeats for different fungal species [52]. Still, we were able to determine the major community shifts in ecological groups and relate these to environmental changes. In addition, our data can be a reflection of changes in host diversity, for example, an increase of plankton parasites through plankton blooms. Molecular approaches that target host-specific fungal taxa and their quantification using qPCR would enable a more detailed understanding of the ecophysiology of fungi in specific habitats that are critical for paleoenvironmental research. Also, studies on the ITS area repeats and the status of DNA degradation would enable future researchers to make stronger conclusions about taxon abundances and therefore community shifts in the context of ecosystem change.

## 5. Conclusions

In summary, our study demonstrates that the ecophysiology of fungi archived into historical sediment records can be used to study the environmental drivers of past communities and processes in both lakes and their surrounding landscapes. We revealed various community changes and related these to paleoproxies and possibly to the changes in host dynamics of host-associated fungi. The observed major community shifts of plankton parasites and mycorrhizal fungi emerging together with increased human activity and climate shifts allow us to infer their susceptibility to environmental factors when most ubiquitous terrestrial groups (e.g., saprotrophs) with wide habitat ranges did not induce extreme community shifts. The effects of temperature, human activity, and landscape openness explained the community changes of fungal groups through plant material transfer and blooms of the planktonic organisms in the lakes. In addition, we highlight that fungi with narrow specificity of substrate or hosts could be more useful as proxies to follow the host and/or substrate dynamics compared with fungi that have broad functionality. A clearer understanding of the links between fungal species presence, abundance, and their host/habitat range over longer time periods needs to be established before we can broadly use ancient fungal communities as independent paleo-variables.

## Figures and Tables

**Figure 1 microorganisms-09-00719-f001:**
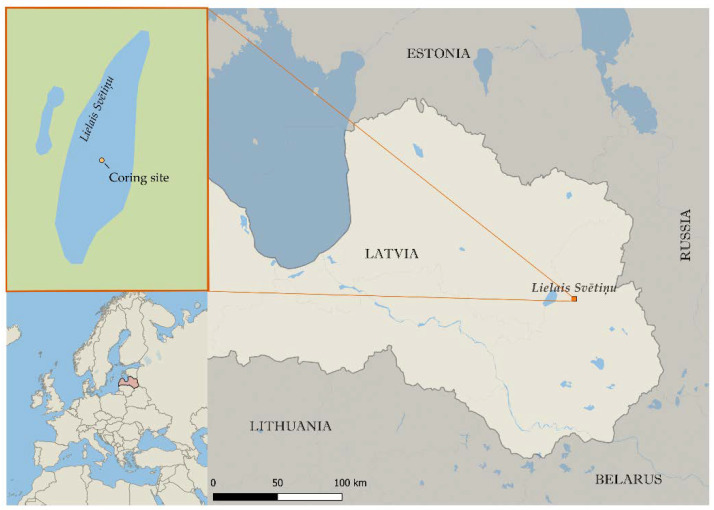
Map showing the location of the lake Lielais Svētiņu and coring site of analyzed sediment cores taken March 2013 in Latvia.

**Figure 2 microorganisms-09-00719-f002:**
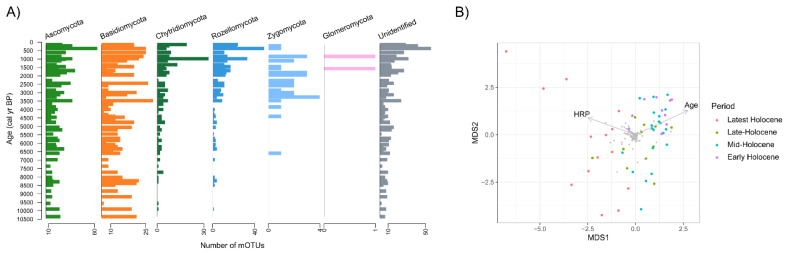
Dynamics of the fungi in the sediment core: (**A**) Richness of observed fungi grouped by phylum; (**B**) Ordination space of PCoA analysis using read abundance (normalized read counts) of mOTUs. Colored points mark various time periods in Holocene and grey points mark the single mOTUs. Arrows indicate linear regression (*envfit* in *vegan* package) of sample age (Age, R^2^ = 0.57, *p* < 0.05) and concentration of human-related pollen (HRP, R^2^ = 0.43, *p* < 0.05) with PCoA space.

**Figure 3 microorganisms-09-00719-f003:**
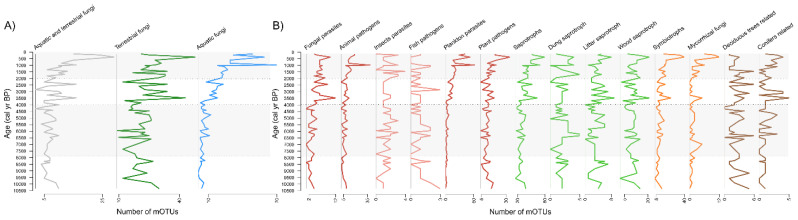
Changes in (**A**) richness of fungi and (**B**) in distributed ecophysiological groups over 10.5 kyr. Dotted lines with colored backgrounds illustrate time periods.

**Figure 4 microorganisms-09-00719-f004:**
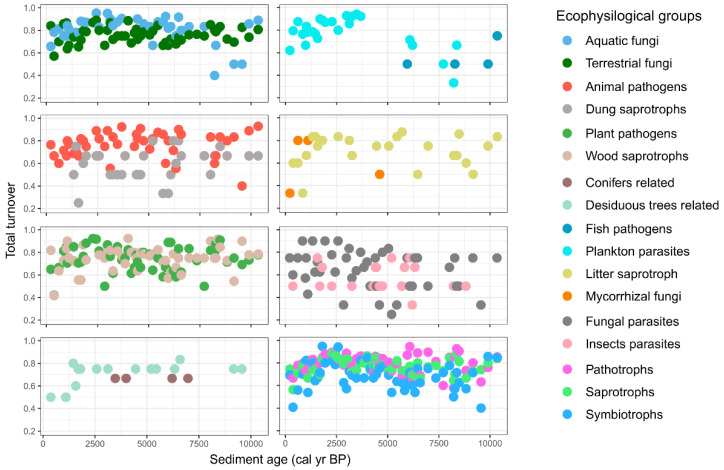
The total species turnover between consecutive sampling (time) points of fungal groups over the last 10.5 kyr. Total species turnover shows the proportion of different species between two time points.

**Figure 5 microorganisms-09-00719-f005:**
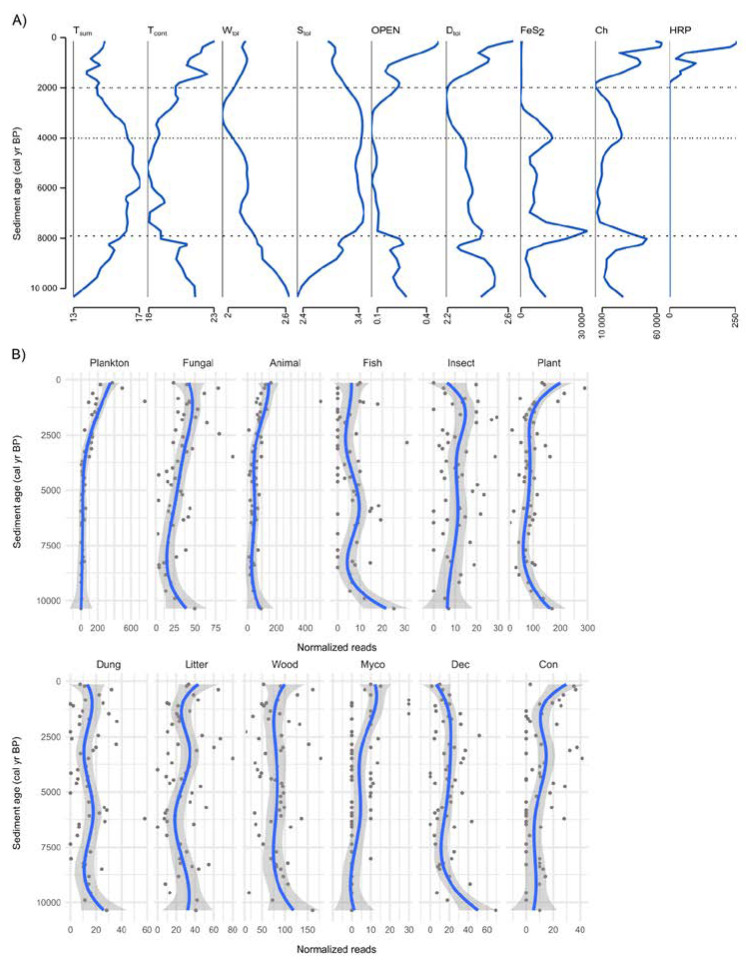
The dynamics of (**A**) environmental paleoproxies and (**B**) ecophysiological groups of fungi over the last 10.5 kyr, modeled using GAM with formula = y ~ splines::bs(x, 5). Assigned ecophysiological roles of fungi abbreviations: plankton parasites (Plankton), animal pathogens (Animal), fish pathogens (Fish), insect parasites (Insect), fungal parasites (Fungal), plant pathogens (Plant), dung saprotrophs (Dung), litter saprotrophs (Litter), wood saprotrophs (Wood), mycorrhizal fungi (Myco), deciduous trees related fungi (Dec), conifers related fungi (Con).

**Figure 6 microorganisms-09-00719-f006:**
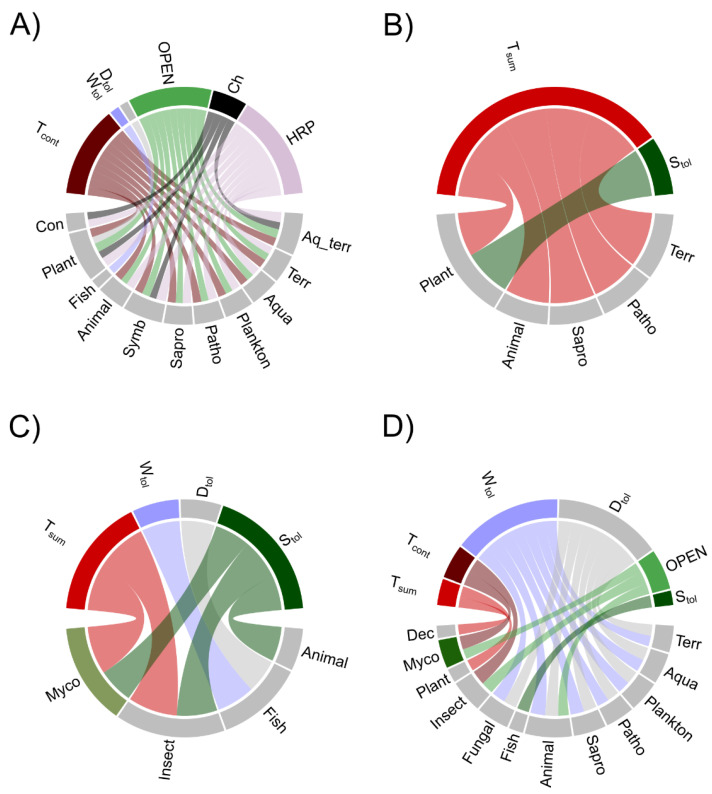
Associations between environmental paleoproxies and abundance in fungal ecophysiological groups over ~10.5 kyr. The two-way Pearson correlations are illustrated using Chord diagrams, where (**A**) shows strong positive correlations (r > 0.6) and (**B**) strong negative correlations (r < −0.6) over the whole timespan (~10.5 kyr); (**C**) shows strong positive correlations (r > 0.6) and (**D**) strong negative correlations (r < −0.6) over the period (2.0–10.5 kyr) excluding possible human impact (last ~2.0 kyr). Assigned abbreviations: water and terrestrial environment fungi (Aq_terr), terrestrial fungi (Terr), aquatic fungi (Aqua), pathotrophs (Patho), saprotrophs (Sapro), symbiotrophs (Symb), plankton parasites (Plankton), animal pathogens (Animal), fish pathogens (Fish), insect parasites (Insect), fungal parasites (Fungal), plant pathogens (Plant), mycorrhizal fungi (Myco), deciduous trees related fungi (Dec), conifers related fungi (Con).

**Table 1 microorganisms-09-00719-t001:** Community change indexes of fungal groups over time.

Ecological Group	Rate Change ^1^	Synchrony ^2^	Variance Ratio ^3^
Water environment	0.04	0.249	16.7
Terrestrial environment	0.013	0.058	3.2
Both environments	0.017	0.121	3.4
Pathotroph	0.038	0.166	15
Animal pathogen	0.017	0.175	4.9
Insects parasite	0.006	−0.166	0.5
Fish pathogen	0.003	−0.276	0.3
Plankton parasite	0.045	0.269	17.8
Fungal parasite	0.009	0.062	1.7
Plant pathogen	0.015	0.077	2.7
Saprotroph	0.025	0.09	6.3
Litter saprotroph	0.004	0.03	1.1
Wood saprotroph	0.009	0.046	1.8
Dung saprotroph	0.004	−0.037	0.7
Symbiotroph	0.018	0.134	5.2
Lichen symbiont	0.0007	−0.326	0.1
Mycorrhizal fungi	0.047	0.141	23.7
Conifer-related fungi	0.001	−0.028	0.7
Deciduous tree-related fungi	0.0002	0.002	1.0

^1^ The *rate change* shows the rate of directional change in community composition over the timespan of the study. ^2^ The *variance ratio* is a positive number ranging from <1 (negative covariance) to >1 (positive covariance) centered at 1 (species vary independently). ^3^ The “gross” *synchrony* values range from −1 (complete asynchrony) to 1 (complete synchrony) with a center value of 0 (species fluctuate independently).

## Data Availability

The sequence data can be found from the SRA database under BioProject accession number PRJNA704193. The bioinformatic filtering scripts are available on Github at https://github.com/liisital/bioinformatic-filtering/blob/main/filtering-script.sh (accessed on 25 March 2021).

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
