# Peer review of "Sedimentary Ancient DNA (sedaDNA) Reveals Fungal Diversity and Environmental Drivers of Community Changes throughout the Holocene in the Present Boreal Lake Lielais Svētiņu (Eastern Latvia)"

_microorganisms, 2021, doi:10.3390/microorganisms9040719_

Round 1

Reviewer 1 Report

The manuscript ID (temporary DOI: https://doi.org/10.3390/xxxxx), entitled `Lake sedimentary ancient DNA reveals fungal diversity and environmental drivers of community changes throughout the Holocene in the present boreal area` by Liisi Talas et al. is focused on temporal dynamics of fungal diversity as revealed by analysis of ancient DNA from boreal lake sediment cores. The authors employed sound methodology and statistical analyses. The findings are solidly supported by a significant number of analysed samples and provide novel information on fungal diversity, its temporal succession and environmental factors controlling the composition and structure of fungal populations during discrete time periods over Holocene. The English is excellent and the message provided to the reader is concise and easy to understand. Overall, this report brings compelling evidence for using sedimentary ancient DNA belonging to aquatic fungi as sensitive proxies for (near) past ecosystem changes in freshwater lakes.

Minor concerns:

  • the title should more precisely indicate the investigated ecosystem (e. `the boreal freshwater Lielais Svētiņu (eastern Latvia)`; the current title suggests a broader survey than it is actually the case; as a suggestion, I found the information associated to BioProject PRJNA704193 (SRA) as a helpful analogue of how title should be written;
  • page 2 –please explain the choice of the sampled lake –what makes this lake distinctive among other boreal freshwater lakes or of (high) scientific interest? This explanation is found in the Discussion part but it would be more helpful for the readers to spot its relevance in the description of the sampling site (either in the Methods or in the Appendix A);
  • page 2 -`…further analysis depending on temperature changes` -add `on the estimated past temperature changes`;
  • Clearer description of the number of samples considered for analysis is needed (what samples/ how many of them have been used for what analysis, see Table S1); please provide details on what `biological replicates` (Table S1) meant to this investigation (as only one core was surveyed, then it is not really clear how biological replicates have been considered);
  • As environmental factors are referred throughout the manuscript (pag 9 onward, Supplem Table S4 and so on), I found no description on whether and how were they measured; using measurements done and reported in other papers must be clearly explained and reasoned;
  • Figure 1 –information on sampling time should be given in the caption.
  • Pag 4 –using `molecular OTU` (mOTU) is quite unusual as most of the readers are familiarized with `OTU` (at cut-off limit of 97%); please explain the choice;
  • 5 – the ` Chytridiomycota` should be consistently written in italics; `Saccharomyces, and Ganoderma`, write `and` in regular fonts;
  • Pag 9, section 3.4 –all abbreviations needs defining; please also see my comments above on the issue of reporting environmental data whose measurements are not explained in the Methods part;
  • Pag 10, Fig. 5 – in the image, the word `Funga` needs correction;
  • Pag 15 - `togethers` need correction;
  • Pag 16 – in ` LSv is a lake with a size of 18.8 ha`, please replace `size` with `area`;
  • Images and data in Supplementary files should be accompanied by captions within the files;
  • Supplementary Table S1, column C –Depth FROM the core

Reviewer 2 Report

Dear Authors

your contribution on fungi from lake sediments is well written, scientifically sound and very interesting. Few comments are in the attached manuscript.
